# The Influence of Workload and Work Flexibility on Work-Life Conflict and the Role of Emotional Exhaustion

**DOI:** 10.3390/bs10110174

**Published:** 2020-11-16

**Authors:** Gabriele Buruck, Anna-Lisa Pfarr, Marlene Penz, Magdalena Wekenborg, Nicole Rothe, Andreas Walther

**Affiliations:** 1Professor for health promotion and prevention, Faculty of Health and Healthcare Sciences, University of Applied Sciences Zwickau, 08056 Zwickau, Germany; anna-lisa.pfarr@gmx.de; 2Institute for Education and Psychology, University of Applied Sciences, 4040 Linz, Austria; marlene.penz@jku.at; 3Institute of Biopsychology, Faculty of Psychology, Technische Universität Dresden, 01069 Dresden, Germany; magdalena.wekenborg@tu-dresden.de (M.W.); nicole.rothe@tu-dresden.de (N.R.); 4Clinical Psychology and Psychotherapy, Department of Psychology, University of Zurich, 8006 Zürich, Switzerland; a.walther@psychologie.uzh.ch

**Keywords:** work-life-conflict, burnout, workload, work flexibility, job demands-resources model

## Abstract

The purpose of this study is to examine the relationship between contextual work-related factors in terms of job demands (workload—**WL**) and job resources (work flexibility—**WF**), work–life conflict (**WLC**) and the burnout dimension emotional exhaustion (**EE**) in a large population-based sample. Building on the job demands resources model (JDRM), we have developed the hypothesis that **WL** has an indirect effect on **EE** that is mediated by **WLC.** We conducted a secondary analysis using data from the Dresden Burnout Study (DBS, *N* = 4246, mean age (SD) = 42.7 years (10.5); 36.4% male). Results from structural equation modelling revealed that **EE** is positively associated with **WL** (β = 0.15, *p* = 0.001) and negatively associated with **WF** (β = −0.13, *p* = 0.001), also after accounting for potential confounding variables (demography, depressive symptoms, and lifetime diagnosis of burnout). Both effects are mediated by **WLC** (*β* = 0.18; *p* = 0.001 and *β* = 0.08; *p* = 0.001, respectively) highlighting the important role of **WLC** in employee health. In summary, **WF** may help to reduce burnout symptoms in employees, whereas **WL** may increase them. Study results suggest that both associations depend on **WLC** levels.

## 1. Introduction

In the rapidly changing working environment of the 21st century, work-related chronic stress is becoming an increasing health risk for Western industrial nations as a whole and is on the rise in many countries [1]. A common consequence of chronic work-related stress is the burnout syndrome [2]. In science and practice, the concept of burnout postulated by Maslach, Schaufeli, and Leiter [3] has gained acceptance as a psychological syndrome induced by work-related stressors, which includes the components emotional exhaustion, depersonalization (e.g., negative attitudes or inner distancing from the work activity) and feelings of reduced performance experience [3,4].

Important research results in the area of possible work-related causes and triggers of burnout and its core component emotional exhaustion (EE) are provided by the job demands resources model (JDRM; [5,6]), according to which both work demands and work-related resources are important factors influencing the health of employees. Quantitative work demands, such as a high workload or time pressure, are among the strongest predictors of emotional exhaustion across multiple studies [7].

The opposite to work requests, described in JDRM, are work resources. They include all those aspects of work that are required to achieve goals, reduce work demands, and associated costs or to promote personal growth and the development of employees [5]. An increasing number of studies point to a direct influence of low levels of work resources on burnout and in particular to emotional exhaustion [8,9,10].

Influence of work on private life also plays an important role in the development of EE through work-life conflicts (WLC; [11,12]). In large heterogeneous population samples examined in Australia and the Netherlands, correlations between these constructs have already been demonstrated [13,14]. However, corresponding studies in the German-speaking population are lacking. We want to begin to fill these gaps by using data of the Dresden Burnout study.

Specifically, the aim of the current study is to investigate the relationships between work demands, work resources, work-life conflict, and the burnout core component emotional exhaustion in a large, heterogeneous sample of German-speaking employees with different occupational backgrounds [15].

### Role of the Work-Life Conflict as Mediator

Several European studies [16,17] as well as international studies, for example, from the USA [18] or Japan [19], have been able to demonstrate across countries and sectors that work-related impairment of private life partly conveys the positive influence of high quantitative work demands on emotional exhaustion. The role of WLC as a mediator between time pressure or a high workload (WL) on the one hand and states of emotional exhaustion on the other has also been confirmed in various longitudinal studies [14,20] and in large, heterogeneous population samples [13,14]. So far, there have only been a few empirical studies that have investigated the health effects of work flexibility (WF) in terms of temporal autonomy. Many publications on JDRM find significant negative correlations between general autonomy [21,22] or experienced control [23] at the workplace and emotional exhaustion. However, the evidence on EE and time margins as a partial aspect of control is limited to a few studies. Nevertheless, the negative relationship between general temporal control or temporal latitude and the impairment of private life is well documented by various studies and meta-analyses. For example, [24,25] found significant negative correlations between the possibility of flexible working hours (schedule control) and WLC.

For this reason, our research model focuses exclusively on job demands such as WL and job resources such as WF and their effect on WLC under control of the depressive symptoms as a condition for emotional exhaustion (see Figure 1). We posit that employees experiencing high-risk job demands (WL) and missing job resources (WF) will suffer work-life-conflicts (WLC) and, in turn, high emotional exhaustion. Put differently, the high-risk work-related factors act as stressors and lead to an imbalance in private life that serves as a condition for burnout. On the other hand, time and content flexibility in the sense of work flexibility can be a resource. Accordingly, the following hypothesis on the effect of job demands on WLC and WL is derived for the present study:
**Hypothesis** **1.**Job demands (WL) have a direct and indirect positive effect on EE mediated by the WLC mediation.

Initial study results indicate that WF partially mediates the negative effect of time margins on emotional exhaustion. For example, a longitudinal study of an Australian population sample [13] found that WLC partially mediated the negative effect of general autonomy at work on the health of employees according to [26]. Yu et al. [27] investigated the possibility of flexible working hours (schedule control), a construct very similar to the time margins defined in this paper. The study also found a significant partial mediation of the negative effect of time flexibility on EE. This is consistent with recent findings, according to which WLC in JDRM also mediates the effect of resources in the motivational process, such as the relationship between control and job satisfaction [28] or between support by the supervisor and job satisfaction [16]. Based on these findings, the following hypotheses on the direct and indirect effects of time margins on EE and WF are derived for the present study:

**Hypothesis** **2.**
*Job resources (WF) have a direct and indirect negative effect on EE mediated by the WLC mediation.*


In addition, various health-related factors have an impact on WLC and EE. For example, [29] found that subjects with a high WLC score exhibited significantly more depressive symptoms in the previous months than those with low WLC. Some studies also show high biological and symptomatic overlap between emotional exhaustion as a component of burnout and depressive symptoms [30,31,32,33]. Others find a strong association between the two constructs [34], which makes the occurrence of emotional exhaustion more likely in depressed subjects. For this reason, the variable depressive symptoms were included as an additional predictor to control potentially conflicting influences (H3, Figure 1). Therefore, we want to test whether H1 and H2 will alter, if a third crucial factor is included.

**Hypothesis** **3.**
*Depressive symptoms have an direct and indirect positive effect on EE mediated by the WLC mediation.*


## 2. Materials and Methods

### 2.1. Procedure and Sample

The aim of the Dresden Burnout Study is to systematically record biological, psychological, and environmental factors in the development of public safety and health services in the entire German-speaking area. Furthermore, its purpose is to investigate the phases of life that accompany them. For a detailed overview of the central questions, design, procedure, and measurements, please refer to the study protocol [15]. For the current article, cross-sectional data of the first survey wave were examined.

Recruitment of participants took place between January 2015 and November 2018 through repeated public relations activities and media coverage in German-speaking countries, including daily newspapers. The study was also promoted on various online platforms and on social media (e.g., Xing, Facebook), and was implemented exclusively through online questionnaires and biomarker analysis. In order to reduce the selection bias and to create a representative sample, an additional 10,000 private households in the city of Dresden were identified and addressed between December 2016 and January 2017. These households were informed about the study procedure and possible participation via mail. The addresses had previously been randomly selected by the residents’ registration office in a randomized drawing. In total 7800 participants registered online for the study, out of which 6490 participants eventually filled in the online questionnaire [15].

The only inclusion criteria for participation in the study were to be aged between 18 and 68 years and to be employed at the time of the study. Furthermore, a sufficient knowledge of German was required to process the questionnaires. The whole sample, included in the following analyses, consisted of 4890 participants (04/2019). Further, we listwise deleted participants with missing values within the independent variables, mediator variable, and dependent variables and further outliers so that the data set was reduced to *N* = 4246. However, the local verification by means of t-tests showed that none of the two model variables WL and WF was missing depending on the criterion variables WLC and EE, so that according to [35] no distortion of the parameter estimation by deleting the cases was to be expected.

A detailed description of the sample, including other demographic information, educational attainment, occupational characteristics, and mental health information, is given in Table 1. The final sample included 2661 women (62.7%) and 1585 men (37.3%) with ages ranging from 18 to 68 years (*M* = 42.71, *SD* = 10.46). The sample corresponds in many areas to a sample recruited at state level from the study of the Federal Institute of Occupational Safety and Health [36]. The large proportion of individuals identified with a high educational level (55.5%) can be described as unusual.

The relatively poor general mental health of the participants is also striking. The proportion of test persons who stated that they had been diagnosed with burnout at some point in their lives was 18.3%, more than four times as high as in the representative German National Health Interview and Examination Survey (DEGS1) [37]. The proportion of participants with clinically relevant depressive symptoms was also higher in the present sample compared to the German general population. Thus, 37.1% of the respondents achieved a score ≥ 10 in the Patient Health Questionnaire (PHQ-9; [38]), which was defined in various studies as the cut-off value for an increased risk of depressive syndrome [38,39]. In comparison, the population-representative DEGS study reported increased risk of depressive syndrome only for 8.1% of the sample (*N* = 7524) [39].

### 2.2. Measurement

After full study information and informed consent, participants were asked to fill out an online questionnaire. The following questionnaires were used for the selected study.

Workload (**WL**) was assessed with the short version of the Effort-Reward-Imbalance Questionnaire (ERI; [40]). The ERI employs a four-point Likert scale ranging from 1 = *I don’t agree at all* to 4 = *I fully agree*. WL is measured with three items, focusing on quantitative aspects of effort, for example: “I have constant time pressure due to a heavy work load”. The internal consistency of the Effort scale has been rated as good in various studies with values of Cronbach’s alpha between α = 0.74 and α = 0.80 [40,41]. Also, in the present study, the internal consistency with a Cronbach’s alpha of α = 0.73 can be considered acceptable according to [42].

Work flexibility (**WF**) was measured with the scale “freedom at work” from the German short version from Copenhagen Psychosocial Questionnaire (COPSOQ, [43]). The scale is composed of four items (the answers are displayed on a five-point Likert scale in steps of 25 with values from 0 = *never* to 100 = *always*). The item values are averaged and a final value range between 0 and 100 is the result. Examples include “If you have private things to do, can you leave your workplace for half an hour without special permission?” or “Can you decide for yourself when to take a break?” The internal consistency of the German scale is higher, with α = 0.78 [44], than the Danish original version (α = 0.68) [45]. The present study received a comparable value for Cronbach’s alpha with α = 0.83, which is considered good [42].

Work-Life Conflict (**WLC**) was likewise measured from COPSOQ with the scale “work privacy conflict”. The scale consists of five items with a five-point Likert scale, for example: “The demands of my work interfere with my private and family life”, and/or “My work takes so much of my time that it has a negative effect on my private life”. The Cronbach’s alpha in the present work (α = 0.90) can be considered good.

Emotional exhaustion (**EE**) was measured with the same-named subscale of an adapted German version of the Maslach Burnout Inventory–General Survey (MBI-GS; [44]), the Maslach Burnout Inventory—General Survey—German Version (MBI-GS-D; [45]). The MBI-GS uses 16 items to measure three dimensions of burnout syndrome: exhaustion, cynicism, and (reduced) professional performance. Only the subscale “emotional exhaustion”, which represents the core symptom of the burnout syndrome, was included in the study [3,45]. The five items of the scale determine how emotionally overwhelmed and exhausted respondents feel due to the demands of the job. The seven Likert-type items of the job-related emotional exhaustion subscale focus on symptoms of fatigue and depleted energy and employ a seven-point frequency rating scale from 0 = never to 6 = daily. An exemplary item is, “Working all day is really a strain for me”. Cronbach’s alpha in the present sample was α = 0.90, which according to [41] indicates good internal consistency.

This coincides with results of other studies, which report values from α = 0.76 to α = 0.86 for both the English [46] and the German version [47] of the EE scale in the MBI-GS.

Depressive symptoms were measured using the Patient Health Questionnaire 9 (PHQ-9, [38]). The PHQ-9 covers nine items on depressive symptomatology. Unlike many other depressiveness questionnaires, PHQ-9 uses each question to capture one of the nine DSM-IV criteria for diagnosing major depression. The Likert-type items of depressive symptoms are presented on a four-point frequency rating scale from 0 = *not all* to 3 = *almost every day*. PHQ-9 is recommended by the DSM-5 working group of the American Psychiatric Association as a tool for measuring the severity of major depression according to the new DSM-5 criteria.

Confounder variables were included in our analysis, such as age, gender, number of children under age of 15 living in the household, working hours per week, length of employment, type of employment (full/part-time), and lifetime diagnosis of burnout, as these were found to have an influence on the proposed relationships in our exploratory analyses.

In addition to the questionnaires used in this study, other questionnaires were used to survey psychological experience and behavior [15].

### 2.3. Statistical Analysis

All computational procedures were performed with the IBM SPSS statistics package (Version 20.0). To test the hypotheses, a structural equation model was calculated in IBM SPSS Amos 22.0. To describe the sample, mean values and standard deviations as well as the percentage distributions of demographic, occupational, and health-related variables were calculated. For the model variables, in addition to mean values and standard deviations, the correlation coefficients were calculated to examine the strength and direction of the relationships between the parameters under investigation. The product-moment correlation according to Pearson used here is also robust to violations of the normal distribution assumption [48].

For the path analysis using observed variables scores and maximum likelihood estimation, we applied the following parameters for evaluating model fit: root mean square error of approximation (RMSEA; good fit: RMSEA < 0.05), standardized root mean square residual (SRMR; good fit: SRMR < 0.10), and comparative fit index (CFI; good fit: CFI > 0.95; cf. [49]). A model with two manifest exogenous and two manifest endogenous variables was tested. Based on the factor depressive symptoms, whether the effect of mediation of WL and WF will change will be analyzed. We therefore tested first a model with four variables and afterwards a model with five variables.

To avoid unequal weighting due to the very different scaling of the two predictors in the interaction term and to avoid multicollinearity effects, the predictors were centered before the interaction term was calculated [50]; see also [51]. To ensure the over identification of the model, the two covariances of the predictors with the interaction term were equated with each other and entered into the model as restrictive parameters [50,52]. The parameters were estimated using the asymptotically distribution-free method (ADF; [53]). With a sample size of *N* = 4246 and a simple model with only four manifest model variables, the present study offered optimal conditions for the application of the ADF method. The overall quality of the model (global fit) was assessed on the basis of the χ² test and selected fit indices. In addition, the significance tests of the model parameters to be estimated and the standardized residuals were used to check the quality of substructures of the model (local fit) [52]. In order to determine the effects of mediation (complete vs. partial mediation), the present study is based on the mediation tests of [54,55,56].

To rule out confounding effects, a number of dichotomous (gender, type of employment, self- reported lifetime diagnosis of burnout) and continuous control variables (age, number of children under age of 15 living in the household, working hours per week, length of employment) were included in the model. For each control variable, direct paths to the endogenous variables as well as covariances with the exogenous variables were modeled (see also [57,58]). To account for possible interdependencies between control variables, they were first included each at a time and then all together into the model. It was observed whether the path coefficients of the model changed significantly when adding the control variables. If this is not the case, confounding effects are not to be expected [57]

## 3. Results

### 3.1. Preparatory Analysis

In total, *N* = 4689 subjects met the abovementioned criteria for participation in the study. After conducting a quality analysis of the data and excluding implausible values, cases with incorrect values and univariate outliers, a final data set of *N* = 4246 participants was obtained and included in the analysis.

The exclusion process of cases during the processing of the data set is shown in Figure 2.

### 3.2. Model Variables

The mean values and standard deviations as well as the intercorrelations of the study variables are shown in Table 2. While the mean values of the subjects for the exogenous variables WL and WF are in the upper third and upper half of their respective scales, respectively, the average scores for the endogenous variables WLC and EE are each very close to the center of the scale. All correlations were significant at the *p* ≤ 0.01 level. The two highest correlations were found between WL and WLC (*r* = 0.46) and between WLC and EE (*r* = 0.53). The correlation between WL and EE was somewhat lower with a correlation coefficient of *r* = 0.37. WF was negatively correlated with all other variables and, with coefficients between *r* = −0.21 with WL and *r* = −0.28 with EE, formed the weakest relationships between the variables.

### 3.3. Assessment and Introduction of the Model

The quality criteria for global model fitting indicate a good overall fit of the model (see Table 3). With χ²(1) = 2715, the result of the Chi-square test lies within the limit of three times the number of degrees of freedom (3 × df) postulated by [35]. The *p*-value (*p* = 0.099) is not significant and thus implies the assumption of the model. This model was aimed at testing H1, H2, and H3. Especially against the background of the large sample and the small number of model parameters, the results of the Chi-square statistics thus indicate a good fit between model and data set.

Figure 3 shows the path model with the standardized path coefficients, the correlations between the exogenous variables and the variance shares of the endogenous variables (R2) resolved by the model. The exogenous variables contained in the model explained 25% of the variance in impairment of privacy and 32% of the variance in emotional exhaustion. All postulated correlations in the model were significant and pointed in the expected direction.

### 3.4. Hypothesis 1: Direct and Indirect Effect from WL on EE via WLC

In hypothesis 1, the claim was made that quantitative work demands also indirectly affect EE by the mediator work-related impairment of private life. Table 4 shows a significant indirect effect of WL on EE (β = 0.18; *p* = 0.001) and thus confirms this hypothesis. The examination of the specific type of mediation effect also confirmed the existence of a partial mediation postulated in hypothesis 1. Thus, the result of the χ² difference test with Δχ² (1) = 93,571 and *p* < 0.001 pointed to a significantly worse model quality of the alternative model 1 without a direct path from WL to EE and consequently indicated a retention of the model with direct effects [58] as well as the partial mediation effect. Hypothesis 1 was therefore accepted.

### 3.5. Hypothesis 2: Direct and Indirect Effect from WF on EE via WLC

In hypothesis 2, it was postulated that WF has an indirect negative effect on EE, mediated by the WLC. This assumption was confirmed by the significant and negative indirect effect of WF on EE in the model (see Table 4, *β* = −0.08; *p* = 0.001). The significant χ² difference test with Δχ² (1) = 97,911 and *p* < 0.001 showed clear differences in model quality between the alternative model 2 without a direct path from WF to EE and the original model, implying that the original model was retained and that there was partial mediation of the effect of WF on RE via the variable WLC. Accordingly, hypothesis 2 was accepted.

### 3.6. Hypothesis 3: Direct and Indirect Effect from Depressive Symptoms on EE via WLC

The integration of depressive symptoms into the model changed the relationships between exogenous and endogenous model variables significantly. All main effects of the variable quantitative WL decreased, whereby in particular the indirect effect on EE via WLC became significantly smaller with a 0.10-point difference in the path coefficient. This was probably due not only to the decrease in the direct effect of WL on WLC, but also due to the significantly reduced effect of WLC on EE. The significant direct and indirect effects of depressive symptoms on WLC and EE are shown in Table 4. The explanation of variance of the endogenous variables was R^2^ = 0.34 for WLC and R^2^ = 0.55 for EE, taking into account the depressive symptoms. The quality of the model remained unchanged when depressive symptoms were included. The correlations of depressive symptoms with the exogenous variables estimated in the model were *r* = 0.27 for the variable WL and *r* = −0.24 for the variable WF. The results therefore provided strong evidence for hypothesis 3.

### 3.7. Integration of Control Variables into the Model

The integration of control variables into the model did not change the relationships between exogenous and endogenous model variables to a significant degree. Neither the path coefficients nor the significance levels of model variables were altered, indicating that confounding effects of control variables on the examined relationships were not to be expected.

## 4. Discussion

The present study provides further evidence on the role of WLC as mediator of emotional exhaustion. In doing so, the present study took into account work demands and work-related resources. Our findings show that the influence of WL and WF on EE was mediated by WLC. Although the correlations for WFC and EE have already been demonstrated in different working conditions [16,17], they have not yet been surveyed in such a large heterogeneous German sample. Our results on the role of WF, however, are an essential addition to the empirical studies, which have so far been scarce. The same applies to clinically relevant pre-existing conditions.

Our first hypothesis investigated the relationship between WL, EE, and WLC, extending previous findings on these constructs [13,20]. By applying the comprehensive and empirically tested framework of the job demand resources model (JDRM), we found a positive association between WL and EE, thusly linking high-risk work conditions to higher EE. A detailed analysis of these relationships using structural equation modelling to control all other variables revealed that only WL played a role in indirectly predicting EE. Similar to WL, WLC shows a comparatively high correlation with EE in meta-analytical studies (*r* = 0.38 to *p* = 0.61) [59], which we could also show in our study (*p* = 0.53).

In hypothesis H1, it was assumed that quantitative work demands such as WL increased EE not only directly, but also indirectly through greater impairment of private life. The results of the structural equation analysis confirmed this expected partial mediation. In accordance with the findings of previous studies and meta-analyses (see z. e.g., [16,17,59]), the impairment of private life had a medium positive effect on the EE of employees (β = 0.43), while the indirect effect of quantitative requirements on EE via WLC, similar to [17], was somewhat smaller with β = 0.18. The findings thus replicate various cross-sectional and longitudinal studies in which the impairment of private life also partly conveyed the negative effects of WL on EE [17,20,60]. They emphasize the important role that conflicts between private and family life play in the development of EE in everyday working life.

Our second hypothesis assumed that the influence of work on private life acts as a mediator between WF and EE. The findings of the present research confirmed the link between WF and EE found by previous studies [11,12]. Furthermore, hypothesis H2 postulated that temporal latitude mitigates EE not only directly but also indirectly by reducing the work-related impairment of private life. As expected, the model found a significant negative, indirect effect between WF and EE, which was slightly higher than in a comparable study (β = −0.08 [27]).

The findings of the present study emphasize the important role played by the balance between work and private life in conveying not only the detrimental effects of job demands but also the alleviating effects of job resources on employee fatigue. They support previous propositions that the impairment of private life acts as a key mechanism linking work conditions to EE and burnout [16,27,28]. As suggested by [16], the present results suggest that WLC should be integrated into the job demands resources model as a central mediator through which both work demands and work resources affect the health of the employees.

Hypothesis 3 assumed the role of depressive symptomatology in this model. Taking depressive symptoms into account, the path coefficients of the effects were reduced by almost half in each case. This is consistent with findings that depressed people report more conflicts between work and private life [29] and at the same time show increased symptoms of EE [32]. Due to the above-average proportion of subjects with substantial depressive symptoms in the present sample, it can be assumed that the correlation between the variables is very high and a smaller effect can be expected for the general population showing significantly lower levels of depressive symptoms. It must certainly be taken into account that the connection the relationship between depressive symptoms and employment status is gendered. Interestingly, men’s mental health is more closely tied to their employment outcomes than is women’s. In contrast, women appear to be more affected by chronic depressive symptoms [61]. In the model including depressive symptoms, the indirect effect of WF on EE was also reduced to β = −0.03, which is similar to the results found by [27]. However, there is substantial empirical evidence that employees, both men and women, who report lack of decision latitude, job strain, and bullying, experience, increased depressive symptoms over time [62]. Certainly, there are still methodological limitations in the recruitment strategy and the selection of the sample to provide a population representative sample. Nevertheless, due to the large sample size comprising of a relatively large proportions of psychologically distressed and healthy individuals, these results can be very important for the promotion of health at work.

### Limitations and Future Directions

Several limitations of the study must be highlighted. First, the secondary analysis examined cross-sectional data, which means that assumptions on causal relationships can be theoretically substantiated but cannot be empirically verified [63]. Thus, the existence of a reverse causal effect cannot be ruled out. In fact, longitudinal studies on WL, WLC, and EE do not only report about the direction of action postulated in this paper, but also reversed causal effects, such as the influence of EE on WLC [64,65], or WL by WLC and EE [66] over time. According to [66], this phenomenon can be explained by so-called “loss spirals”; according to [67], employees whose energy resources are already exhausted are less able to master further challenges, which can lead to an accumulation or intensification of conflicts and demands. Still other studies indicate that neither one nor the other direction of effects dominates, but that workload, work-family conflict, and work flexibility simultaneously influence each other (so-called reciprocal effects; [20,66]). Although the Dresden Burnout Study (DBS) is performed longitudinally, only a cross-sectional analysis was performed for this study, as no longitudinal data were available at the time of analysis.

A second shortcoming concerns the performed analysis in the study. A path analysis with observed variables was chosen to test the hypotheses. However, the large study sample would have allowed testing of two models of structural equations with latent variables, comparing both partial and full mediation models using the Bayesian Information Criterion, which would have been an even more powerful statistical analysis.

Third, the analysed data were collected exclusively by means of self-assessment questionnaires. Thus, the simultaneous assessment of predictor and outcome by the same person may lead to the relationships under investigation being overestimated by artefacts such as common method variance or evaluative bias [68]. In addition, self-reports, especially in people with depressive symptoms, carry the risk of systematically misjudging work characteristics and thus overestimating the effects of demands and resources on health in the overall sample [69,70]. Although the depressive symptoms were controlled in the present study, a systematic common method bias cannot be ruled out in the present study either.

A fourth limitation of our study concerns the lack of representativeness of the sample for the German general population, which limits the generalizability of the results to German employees. For example, in comparison to other representative surveys of the German population, the participants in the Dresden Burnout Study not only had a higher level of education, but also worked significantly more hours per week and had prevalence rates for the lifetime diagnosis of burnout and depressive symptoms that were more than four times as high in each case [37,39]. These results are hardly surprising, since a higher proportion of subjects who are affected by the subject matter of the study and are therefore particularly interested in participating can be expected in an occasional sample. Although the sample matched the German population in many characteristics relevant to the research question and deviating characteristics were controlled in the present sample, a transfer of the results to the entire German workforce cannot be made without reservations.

Finally, we would like to point out a limitation of the study with regard to the completeness of the examined construct of the influence of work on private life. In contrast, the role of problems and conflicts spilling over from private life into work [71,72] on the effect of WL and WF and on EE was not considered. Although WLC is generally more associated with health outcomes than private life into work [73], it is nevertheless conceivable that private conflicts caused by the impairment of private life could in turn have a negative impact on the working lives of employees (so-called reciprocal effects; see [61]) and thus further fuel the development of exhaustion. Due to the lack of consideration or control of LWI in the present study, it remains unclear to what extent the identified effects between the impairment of private life and EE are exclusive to them.

Although the effects examined in the model showed very high levels of significance almost without exception, which was also due to the large sample, it is possible to clearly distinguish between significant and less significant correlations in direct comparison based on the level of the path coefficients. The results show that work-related impairment of private life plays an important role in the development of emotional exhaustion in the everyday working lives of German-speaking employees. The findings may reflect the current trend in the world of work, according to which private goals and a balance between work and private life are becoming increasingly important for employees (so-called non-work orientations; see [74]). This could lead to a situation in which employees perceive and evaluate a restriction of their private life and the associated costs, such as emotional exhaustion, as more negative.

Despite methodical and content wise limitations and based on the results specified above, the study makes a significant contribution to the clarification of the association of work factors and emotional exhaustion as one burnout characteristic in the German-speaking countries. It is the first study to prove that the established relationships between quantitative work demands, the impairment of private life, and emotional exhaustion are to be found across professions in Germany. Following large population studies in Australia and the Netherlands [13,14], a comparable study of the German working population is now available, which confirms the negative effects of WL on EE and underlines the important role of WLC in communicating these effects. In addition, the study identified WF as an effective work resource, which has so far received little attention in research, especially in connection with emotional exhaustion.

Due to the cross-sectional design, statements on causality were not possible in the present study. Future studies should therefore examine the determined effects in a similar setting again in a longitudinal way. In particular, there have hardly been any longitudinal or intervention studies on the effects of informal time margins in relation to both WLC and EE [75]. A first step in this direction is the work of [76], in which women who were able to leave their jobs for a few hours without problems reported fewer WLCs six months later. The study by [77] also found positive longitudinal effects of flexitime for women in the UK. This contributes to our understanding of flexible working not only as a tool for work–life balance, but also as a tool to enhance and maintain individuals’ work capacities in periods of increased family demands. With regard to WL, WLC, and EE, diary studies (see e.g., [78]) in addition to further longitudinal studies are needed to investigate causal relationships further. Diary studies collect the parameters of interest several times a day using methods such as experience sampling via smartphones, which could help to detect causal relationships in real time and thus to better understand possible reciprocal and simultaneous effects.

In a future analysis of the effects, large population samples broken down by occupational groups would also be worthwhile. This would ensure that the identified correlations between job characteristics, WLC, and EE are indeed valid across occupations and would also allow a comparison of different occupational metiers using the same methodological approach. Similar to the present paper, previous large survey studies on the relationship between WL, WLC, and EE have lacked such an approach [13,14].

Initially, prevention measures should be aimed at reducing quantitative work demands such as time pressure, interruptions in the workplace, or excessive workloads, as these have been shown in the present study to be predictive of an increase in EE among employees (see also [79]). A direct reduction in workload at the level of ratio prevention could be achieved most easily by reducing the amount of work, for example, by reducing the scope of services, or by adequately staffing the system [79]. Furthermore, it is recommended that employees will be relieved by optimizing task and work processes, such as improved processes and avoiding disruptions and interruptions [80]. Agreeing on realistic targets and performance standards can also be an effective way of reducing the burden of work flexibility in companies [81].

With regard to work organization, flexible work arrangements, such as formal or informal leeway of working time (flexitime) and place of work (flexplace) play an important role in limiting WLC [82]. Formal offers by companies include, for example, telecommuting or mobile working, flexible working time models such as flexitime, or the possibility of working overtime [77].

With regard to work flexibility, the present paper was able to show, in line with previous studies [75,83], that the possibility of spontaneously taking a day off or leaving the workplace for a few hours, as well as flexible break arrangements, led to a significant reduction in work–private life conflicts. In addition, as this study showed, informal time off also contributed to a direct reduction of emotional exhaustion at the workplace, which means that it ought to be considered in work design at companies. In particular, effective break design, for example, by enabling informal short breaks during working hours, can lead to a significant reduction in negative health outcomes among employees [83]. According to previous findings, it is evident that the work-family conflict is primarily due to exhaustion and people being unable to handle their workload. This in itself is an important field of research and area of interest. As a consequence, an elimination of work-related stress and pressure is highly recommended [84]. Despite the strong statistical correlation between emotional exhaustion and depression [31,32] as well as depression and work-demands [85], this was not considered as a possible constitutive variable in the effect of work characteristics on Emotional Exhaustion (EE) in this study. Even if one assumes that burnout, and in particular emotional exhaustion, is part of depressive symptoms and less of an independent syndrome [86,87], a predictor of depressive symptoms in studies nevertheless makes sense, as depressive subjects often systematically over- or underestimate working characteristics in subjective rating scales [85]. The methodological strength of the study, besides the size and heterogeneity of the sample, which in many respects corresponded with representative data from the German general population, is the integration of health-related measures and especially depressive symptoms as crucial determining factors into the model.

## 5. Conclusions

To sum up, one can say that the results of the structural equation analysis showed, in accordance with the postulated assumptions, a direct and indirect positive effect of quantitative requirements, a direct and indirect negative effect of temporal margins, and a direct and indirect positive effect of depressive symptoms on EE. The mediator variable impairment of the private life thereby partly mediated the effect of the predictors and showed even the strongest effect on EE. This study makes an important contribution to elucidating the mechanisms that lead to burnout of German-speaking employees. It replicates statements of the job demand resources model (JDMR), which illustrates that job demands are an important predictor of EE. The high proportion of people with high depressive symptomatology needs to be taken into account when interpreting the results. Despite some methodological shortcomings, the present study provides an important contribution to health-promoting work design highlighting the importance of WF.

## Figures and Tables

**Figure 1 behavsci-10-00174-f001:**
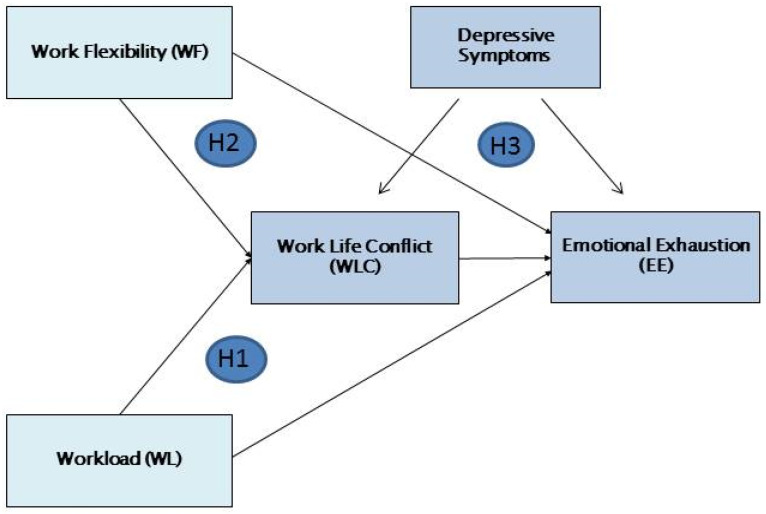
Schematic illustration of the investigated research model with its three hypotheses (H1–H3).

**Figure 2 behavsci-10-00174-f002:**
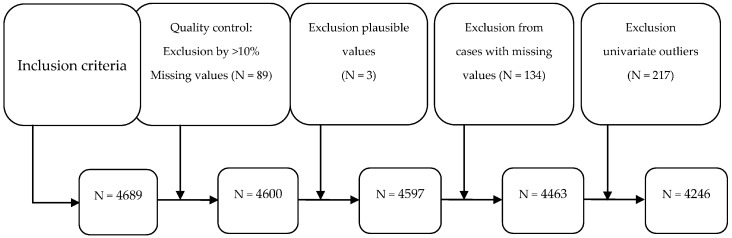
Sample flow during the exclusion process.

**Figure 3 behavsci-10-00174-f003:**
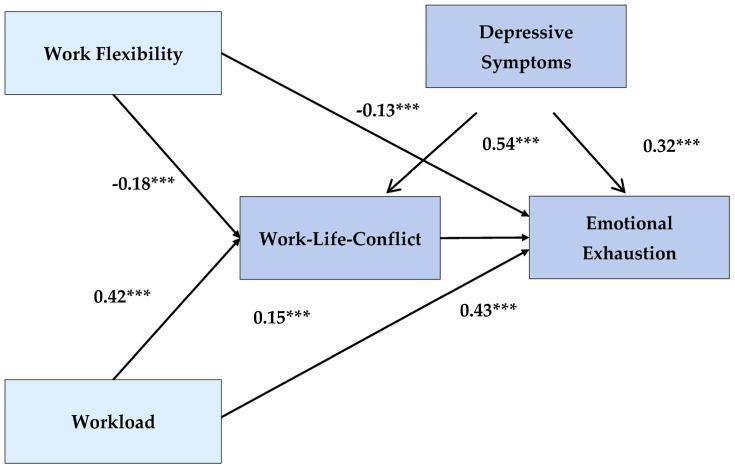
Path model with standardized coefficients, covariances, and their respective significance levels (*** *p* ≤ 0.001) and with the resolved variances of the endogenous variables (R²).

**Table 1 behavsci-10-00174-t001:** Description of the sample.

	German-Speaking Working Adult Participants (*N* = 4246)	
	M (SD)	Scale
Demographics and Education		
Age	42.71 (10.46)	18.0–68.0
Women (%)	62.7	
One or more children in the household (%) ^a^	28.6	
University degree (%)	55.4	
Job-related characteristics		
Full-time (%)	79.8	
Working Hours per Week ^b^	41.87 (7.08)	21.0–62.0
Duration of employment (Years)	9.5 (9.23)	0.0–46.0
Shiftwork (%)	13.9	
Occupational Status (%)		
Employees	85.2	
Freelancer	6.1	
Civil servants	8.7	
Job requirements (%)		
Mostly mental	81.3	
Mostly physical	1.8	
Combined	16.9	
Mental Health Conditions		
Self-reported Lifetime Diagnosed Burnout (%)	18.3	
Depressive Symptoms ^c^	8.36 (5.37)	0.0–27.0

^a^ Children were under 15 years old. ^b^ Including overtime. ^c^ Patient Health Questionnaire (PHQ-9).

**Table 2 behavsci-10-00174-t002:** Means, standard deviations, internal consistencies, and intercorrelations of model variables.

	Variables	*α*	*M (SD)*	*Scale*	1	2	3	4
1	Workload (WL)	0.73	9.37 (1.94)	3–12	1			
2	Work-life conflict (WLC)	0.90	51.57 (25.66)	0–100	0.46 **	1		
3	Work flexibility (WF)	0.83	65.72 (25.13)	0–100	−0.21 **	−0.27 **	1	
4	Emotional exhaustion (EE)	0.90	3.08 (1.52)	0–6	0.37 **	0.53 **	−0.28 **	1
5	Depressive Symptoms	0.91	8.36 (5.37)	0–3	0.28 **	0.44 **	−0.24 **	

Notes: *α* = Cronbachs Alpha, *M* = Mean, *SD* = Standard Devision, *r* = Correlation coefficient Pearson. ** *p* ≤ 0.01, *N* = 4246.

**Table 3 behavsci-10-00174-t003:** Quality criteria of the model adaptation.

χ^2^	*df*	*p*	RMSEA (PCLOSE)	SRMR	CFI	TLI	IFI
2.715	1	0.099	0.020 (0.946)	0.0117	0.999	0.989	0.999

Notes: *df*: Degree of Freedom, RMSEA: Root-Mean-Square-Error-of-Approximation, PCLOSE: Probability of Close Fit for the RMSEA, SRMR: Standardized Root Mean Square Residual, CFI: Comparative Fit Index, TLI: Tucker-Lewis-Index, IFI: Incremental Fit Index. *N* = 4246.

**Table 4 behavsci-10-00174-t004:** Comparison of standardized path coefficients with and without depressive symptoms.

	Emotional Exhaustion		Work-Life Conflict
	Direct Effects	Indirect Effects	Direct Effects
	β	β	β
**Main effects**			
Workload	0.15 *** **0.11 *****	0.18 *** **0.08 *****	0.43 *** **0.35 *****
Work Flexibility	−0.13 *** −**0.07 *****	−0.08 *** −**0.03 *****	−0.18 *** −**0.12 *****
Work-Life Conflict	0.43 *** **0.23 *****		
Depressive Symptoms	**0.54 *** (0.52−0.56)**	**0.07 *** (0.06−0.08)**	**0.32 *** (0.30−0.35)**

Notes: β: Standardized direct or indirect effect in the original model (recte) or taking into account the control variable depressive symptoms (*italics*, **bold**); n. s.: effect is not significant; * *p* ≤ 0.05, ** *p* ≤ 0.01, *** *p* ≤ 0.001; *N* = 4246.

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
