# Peer review of "The Influence of Workload and Work Flexibility on Work-Life Conflict and the Role of Emotional Exhaustion"

_behavsci, 2020, doi:10.3390/bs10110174_

Round 1

Reviewer 1 Report

Dear authors, it was a pleasure to read your manuscript. The study is quite impressive and conducted on a very large German sample. Despite I think that your study is quite interesting I have some suggestions that I hope you would find helpful in improving the paper:

  • I suggest to always use the same names when you refer to the variables. For example hypothesis 3 is quite confusing. I would add that when you refer to job demands and job resources you put in parentheses the contructs you measured. The same I think it is needed to be done for work-related impairments.Hypothesis 3 it is not clear, what does it mean reduce the main effects? It look like these are three hypotheses together.
  • This sentence “it is also to be expected that individuals who have already depicted burnout symptoms once in their lives are at a higher risk for the recurrence of emotional exhaustion doesn’t look very clear as emotional exhaustion is a component of burnout.
  • One of the main doubts I have refer to depressive symptoms. How can they be an antecedent of WLC and emotional exhaustion? The literature I am aware of, would suggest them as an health outcome of EE and not an antecedent. I think you should provide a more in depth argument for describing why depressive symptoms are considered as a mediator.
  • You state that “The only inclusion criteria for participation in the study were for potential participants to be in the 18 to 68 years age group.” Maybe another criteria was also that they were working? Otherwise it is difficult to measure the variables you mentioned
  • You correctly mentioned that it was surprising the proportion of people who declare they have been diagnosed with burnout at some point and consequently also the proportion of people with depressive symptoms. I think this is worthy to be included in the paper but at the same time you should mention in the discussion section, why you think you have these data (maybe the recruitment strategy) and if/how this could have affected your results.
  • Can you add a sample item both for work flexibility and work on private life? There is only one item and it is not clear to wchich scale it refer (it looks it refer to work flexibility but form how the sentence is structured it looks like it is an item wor WLC. Also, it would be useful to divide the two constructs and not refer to them together in the measurement section.
  • In section 3.5. you use the acronym RE. Was it EE? Also now Work family conflict is WLFC and no more WLC. Please be consistent throughout the manuscript
  • It is not clear to me why you didn’t run only one model. My suggestion is that you run only one model with depressive symptoms as the outcome variable of emotional exhaustion as also the theoretical argument about depressive symptoms affecting work-life conflict and emotional exhaustion it is not convincing so far

Minor comments:

  • “In science and practice, the concept of burnout postulatedby (Maslach, Schaufeli, and Leiter 2001)” you probably should report the year in parentheses,
  • “JDRM” usally the model is cited as JD-R model
  • Deprisse symptoms is the only variable without an acronym, for the sake of clarity maybe you should use an acronym also for it
  • Sometimes in the tables you use 3 decimals and sometimes 2. I think it is better to decide and stick to one of the two.

Author Response

Reviewer 1

RP1.1: I suggest to always use the same names when you refer to the variables. For example hypothesis 3 is quite confusing. I would add that when you refer to job demands and job resources you put in parentheses the contructs you measured. The same I think it is needed to be done for work-related impairments. Hypothesis 3 it is not clear, what does it mean reduce the main effects? It looks like these are three hypotheses together. One of the main doubts I have is referring to depressive symptoms. How can they be an antecedent of WLC and emotional exhaustion? The literature I am aware of, would suggest them as a health outcome of EE and not an antecedent. I think you should provide a more in depth argument for describing why depressive symptoms are considered as a mediator.

AR1.1: We thank Reviewer 1 for this point and apologize for not being concise enough in our writing. We have checked and uniformly adjusted all variable names. Furthermore, we have revised all hypotheses - especially hypothesis 3 - and made them more precise in terms of concepts and content.

RP1.2: This sentence “it is also to be expected that individuals who have already depicted burnout symptoms once in their lives are at a higher risk for the recurrence of emotional exhaustion” doesn’t look very clear as emotional exhaustion is a component of burnout.

AR1.2: We appreciate this helpful comment. We have now revised the manuscript by identifying emotional exhaustion as one dimension of burnout also linguistically.

RP1.3: You state that “The only inclusion criteria for participation in the study were for potential participants to be in the 18 to 68 years age group.” Maybe another criteria was also that they were working? Otherwise it is difficult to measure the variables you mentioned

AR1.3: Thank you for highlighting this. In the methods section, we have added the inclusion of employment in lines 134/135. In the preparation of the data analysis, we excluded participants who self-reported to be not employed in the survey.

RP1.4: You correctly mentioned that it was surprising the proportion of people who declare they have been diagnosed with burnout at some point and consequently also the proportion of people with depressive symptoms. I think this is worthy to be included in the paper but at the same time you should mention in the discussion section, why you think you have these data (maybe the recruitment strategy) and if/how this could have affected your results.

AR1.4:  Thank you very much for those helpful recommendations. Since the Dresden Burnout Study explicitly examines burnout and potential risk factors, the interest of affected people to participate was naturally great. Nevertheless, we tried to recruit a population representative sample by inviting 10'000 randomly selected individuals from a population registry from the city of Dresden. However, the goal of achieving a population representative sample did not entirely succeed as observable in the disproportionately high proportion of women in the study. Nevertheless, these results can be very important for the health-promotion of work. We have supplemented this information in the methods section and the discussion section in lines 397/398.

RP1.5:  Can you add a sample item both for work flexibility and work on private life? There is only one item and it is not clear to which scale it refer (it looks it refer to work flexibility but form how the sentence is structured it looks like it is an item wor WLC. Also, it would be useful to divide the two constructs and not refer to them together in the measurement section.

AR1.5: Thank you for this valuable comment. We have no included a sample item in the methods section.    

RP1.6: In section 3.5. you use the acronym RE. Was it EE? Also now Work family conflict is WLFC and no more WLC. Please be consistent throughout the manuscript

AR1.6: Thank you for highlighting this mistake. We have revised the document being now consistent with the use of abbreviations with regard to the constructs used.

RP1.7: It is not clear to me why you didn’t run only one model. My suggestion is that you run only one model with depressive symptoms as the outcome variable of emotional exhaustion as also the theoretical argument about depressive symptoms affecting work-life conflict and emotional exhaustion it is not convincing so far.

AR1.7: We thank the reviewer for his comments. We have clarified the mistake linguistically (line 275). We have tested depressive symptoms as a predictor and not as a mediator.

Reviewer 2 Report

The article is well written and I find it interesting. It has a good theoretical basis that supports the hypotheses formulated and the model proposed.

However, the analyses performed can be improved. The authors have proposed a path analysis with observed variables to test their hypotheses, using the total scores in each of the questionnaires. That could be very appropriate if they had a small sample size or many measuring instruments with many items. But they have a sample of more than 4000 participants, which would allow them to test two models of structural equations with latent variables. This would be a more powerful analysis approach and would allow them to compare both partial and full mediation models using the Bayesian Information Criterion (BIC), with lower values indicating better fit.

Before testing both models, a confirmatory factor analysis should be carried out to confirm the unifactorial structure of each measurement instrument, offering the fit indices and the Composite Reliability Index in each case. In order to evaluate the fit of the models, the indices usually recommended by the specialized literature can be used for this type of estimation: CFI, RMSEA, SRMR, and BIC for comparing both models. MLR estimation can be used. The presentation and comment on the results are much simpler. A table could show the fit indices of both models, including the BIC.

On the other hand, I don't understand what role the control variables play. The authors indicate that they include them in the analysis, but do not include them in the models. They are only used for exploratory preliminary analysis, but they are not found anywhere. If these variables are relevant, why not include them in the model? The only thing to do when considering them is to indicate the scale of measurement of each one (number of children would be continuous, and type of work dichotomous, for example). But in this case, they would need to justify why these variables are relevant and how they affect the other variables included in the model.

The bibliography does not follow the format established by the journal. If you use a reference manager such as Mendeley or Refworks, you can download it from Zotero, as indicated in the instructions for authors, and import it: https://www.zotero.org/styles/?q=id%3Amultidisciplinary-digital-publishing -institute.

As a minor issue, on line 96 I think the authors mean "In accordance with these findings ...", and on line 180 the authors describe the 5-point Likert scale from 0 to 100.

Author Response

RP2.1: However, the analyses performed can be improved. The authors have proposed a path analysis with observed variables to test their hypotheses, using the total scores in each of the questionnaires. That could be very appropriate if they had a small sample size or many measuring instruments with many items. But they have a sample of more than 4000 participants, which would allow them to test two models of structural equations with latent variables. This would be a more powerful analysis approach and would allow them to compare both partial and full mediation models using the Bayesian Information Criterion (BIC), with lower values indicating better fit.

Before testing both models, a confirmatory factor analysis should be carried out to confirm the unifactorial structure of each measurement instrument, offering the fit indices and the Composite Reliability Index in each case. In order to evaluate the fit of the models, the indices usually recommended by the specialized literature can be used for this type of estimation: CFI, RMSEA, SRMR, and BIC for comparing both models. MLR estimation can be used. The presentation and comment on the results are much simpler. A table could show the fit indices of both models, including the BIC.

AR2.1: Thank you for your feedback. We agree that testing two models with latent variables including a confirmatory factor analysis would be a more powerful statistical approach for our study. However, a model with manifest variables as the simpler and straighter forward analysis better fit our team’s resources and timeframe and was considered appropriate for the scope of the paper. To evaluate the fit of the models, we used the indices as recommended (CFI, RMSEA, SRMR). Comparing two alternative models with the χ² difference test to determine the effects of mediation is also an approach commonly used by authors in SEM (Kelloway, 1998, see also Arnold, Turner, Barling, Kelloway, and McKee, 2007; Zacher, Jimmieson, and Bordia, 2014).

With regard to an additional CFA, we only used established questionnaires such as the ERI or the MBI to measure variables. The unifactorial structure of the scales in these measures has been confirmed in various studies (e.g. Li et al., 2012; Nübling et al., 2006; Bakker, Demerouti, & Schaufeli, 2002; Schutte, Toppinen, Kalimo, & Schaufeli, 2000), which is why we did not carry out a CFA in our study. To ensure a good reliability of the scales, we calculated Cronbachs alpha as a common index for composite reliability. 

To acknowledge the advantages of testing a model with latent variables, we will add the following statement to the discussion (line 415).

“A second shortcoming concerns the performed analysis in the study. A path analysis with observed variables was chosen to test the hypotheses. However, the large study sample would have allowed testing two models of structural equations with latent variables, comparing both partial and full mediation models using the Bayesian Information Criterion, which would have been an even more powerful statistical analysis.”

RP2.1: On the other hand, I don't understand what role the control variables play. The authors indicate that they include them in the analysis, but do not include them in the models. They are only used for exploratory preliminary analysis, but they are not found anywhere. If these variables are relevant, why not include them in the model? The only thing to do when considering them is to indicate the scale of measurement of each one (number of children would be continuous, and type of work dichotomous, for example). But in this case, they would need to justify why these variables are relevant and how they affect the other variables included in the model.

AR 2.2: Thank you for pointing out this shortcoming. We included all control variables mentioned in line 209 in the model to test for confounding effects. However, this was maybe not sufficiently described in our methods, which is why we added the following paragraph to the paper (line 241):

“To rule out confounding effects, a number of dichotomous (gender, type of employment, lifetime diagnosis of burnout ) and continuous control variables (age, gender, number of children under age of 15 living in the household, working hours per week, length of employment) were included in the model. For each control variable, direct paths to the endogenous variables as well as covariances with the exogenous variables were modelled (see also Baeriswyl, Krause, Elfering, & Berset, 2017; Baeriswyl et al., 2016). To account for possible interdependencies between control variables, they were first included each at a time and then all together into the model. It was observed whether the path coefficients of the model changed significantly when adding the control variables. If this was not the case, confounding effects were not to be expected (Baeriswyl et al., 2017). “  

We also added a separate paragraph to report on the results of integrating the control variables in the model (line 338) to provide a better understanding for the reader:

3.7. Integration of control variables into the model

            The integration of control variables into the model did not change the relationships between exogenous and endogenous model variables to a significant degree. Neither the path coefficients nor the significance levels of model variables were altered, indicating that confounding effects of control variables on the examined relationships were not to be expected.

As none of the control variables significantly altered the relationships of endogenous and exogenous variables in our model, we refrained from providing more information on the analysis such as reporting path coefficients or model fit.

RP2.3: The bibliography does not follow the format established by the journal. If you use a reference manager such as Mendeley or Refworks, you can download it from Zotero, as indicated in the instructions for authors, and import it: https://www.zotero.org/styles/?q=id%3Amultidisciplinary-digital-publishing -institute.

AR2.3: Thank you for pointing out this shortcoming. We have revised the entire bibliography according to the guidelines of the journal.

RP2.4: As a minor issue, on line 96 I think the authors mean "In accordance with these findings ...", and on line 180 the authors describe the 5-point Likert scale from 0 to 100.

AR2.4: Thank you for highlighting this. We have revised the respective sections.

New References Li. J., Loerbroks, A., Shang, L., Wege, N., Wahrendorf, M. & Siegrist, J. (2012b). Psychometric properies and differential explanation of a short measure of effort-reward-imbalance at work: A study of industrial worker in Germany. American Journal of Industrial Medicine. 55, 808-815.

Reviewer 3 Report

The paper approaches the very current topic, the relationship between work-related factors and burnout. Although the paper has reached a very good level, before publication, there are several aspects that need to be addressed, namely:

  • please, clarify the identified gap which you are aiming at fulfilling through your article;
  • in the Introduction part of the article, please, add the text when explaining Hypothesis 3 (what can be seen from Figure 1);
  • Instead of "We therefore assume that hypothesis 1 is true." write Hypothesis 1 is accepted.
  • In conclusions, it is usually one paragraph that simply and succinctly restates the main ideas and arguments, pulling everything together. You write only the contribution of your research in it. Please, reconstruct the conclusion, also by adding limitations from discussion part of your article.
  • Please consider verifying the language, there are some grammar/writing errors. (the comma is missing for example - "We, therefore, assume that hypothesis 1 is true."

Thank you for the opportunity of reviewing your interesting article. It addresses a topic which is within the journal s scope and uses relevant literature to perform the content analysis.

Author Response

RP3.1: please, clarify the identified gap which you are aiming at fulfilling through your article;

AR3.1: Thank you very much for pointing this out. In the introduction, we have now specified what contribution our study provides with regard to the research gap (line 52,53). 

RP3.2: in the Introduction part of the article, please, add the text when explaining Hypothesis 3 (what can be seen from Figure 1);

 AR3.2: We thank the reviewer for this comment. We revised the Introduction section with regard to the explanation of Hypothesis 3 as suggested. 

RP3.3: Instead of "We therefore assume that hypothesis 1 is true." write Hypothesis 1 is accepted. Please consider verifying the language, there are some grammar/writing errors. (the comma is missing for example - "We, therefore, assume that hypothesis 1 is true."

AR3.3: Thank you very much for the comment. We revised the entire manuscript for grammar and spelling.

RP3.4: In conclusions, it is usually one paragraph that simply and succinctly restates the main ideas and arguments, pulling everything together. You write only the contribution of your research in it. Please, reconstruct the conclusion, also by adding limitations from discussion part of your article.

AR3.4: Thank you for highlighting this. The conclusion now sums up the main ideas, findings, and arguments at the end of the manuscript.

Round 2

Reviewer 1 Report

Thanks to the authors for their revision of the manuscript. The information given are clear and improve significantly the paper.

Some of my concerns have been adequately addressed.

Reviewer 2 Report

I think the article has improved after the reviews made and the reading is easier. I still think that it would have improved using a structural equation model with latent variables, although it is not wrong to use a path analysis.